# Raman Spectroscopic Analysis to Detect Reduced Bone Quality after Sciatic Neurectomy in Mice

**DOI:** 10.3390/molecules23123081

**Published:** 2018-11-25

**Authors:** Yasumitsu Ishimaru, Yusuke Oshima, Yuuki Imai, Tadahiro Iimura, Sota Takanezawa, Kazunori Hino, Hiromasa Miura

**Affiliations:** 1Department of Bone and Joint Surgery, Ehime University Graduate School of Medicine, Shitsukawa, Toon 791-0295, Ehime, Japan; marup_59@yahoo.co.jp (Y.I.); kazunori330225@yahoo.co.jp (K.H.); miura@m.ehime-u.ac.jp (H.M.); 2Biomedical Optics Laboratory, Graduate School of Biomedical Engineering Tohoku University, Aramaki Aza Aoba, Aoba-ku, Sendai 980-8579, Miyagi, Japan; 3Department of Gastroenterological and Pediatric Surgery, Oita University Faculty of Medicine, Idaigaoka, Hasama-machi, Yufu City 879-5593, Oita, Japan; 4Oral-Maxillofacial Surgery and Orthodontics, University of Tokyo Hospital, 7-3-1 Hongo, Bunkyo-ku 113-8655, Tokyo, Japan; 5Division of Integrative Pathophysiology, Proteo-Science Center, Ehime University, Shitsukawa, Toon 791-0295, Ehime, Japan; y-imai@m.ehime-u.ac.jp; 6Division of Bio-imaging, Proteo-Science Center, Ehime university graduate school of medicine, Shitsukawa, Toon 791-0295, Ehime, Japan; iimura@m.ehime-u.ac.jp; 7Division of Analytical Bio-Medicine, Advanced Research Support Center, Ehime University, Shitsukawa, Toon 791-0295, Ehime, Japan; 8Molecular Medicine for Pathogenesis, Ehime university graduate school of medicine, Shitsukawa, Toon 791-0295, Ehime, Japan; s_takane@m.ehime-u.ac.jp

**Keywords:** osteoporosis, bone quality, Raman spectroscopy, sciatic neurectomy, hydroxyapatite, mineral maturity, bone matrix, collagen crosslink

## Abstract

Bone mineral density (BMD) is a commonly used diagnostic indicator for bone fracture risk in osteoporosis. Along with low BMD, bone fragility accounts for reduced bone quality in addition to low BMD, but there is no diagnostic method to directly assess the bone quality. In this study, we investigated changes in bone quality using the Raman spectroscopic technique. Sciatic neurectomy (NX) was performed in male C57/BL6J mice (NX group) as a model of disuse osteoporosis, and sham surgery was used as an experimental control (Sham group). Eight months after surgery, we acquired Raman spectral data from the anterior cortical surface of the proximal tibia. We also performed a BMD measurement and micro-CT measurement to investigate the pathogenesis of osteoporosis. Quantitative analysis based on the Raman peak intensities showed that the carbonate/phosphate ratio and the mineral/matrix ratio were significantly higher in the NX group than in the Sham group. There was direct evidence of alterations in the mineral content associated with mechanical properties of bone. To fully understand the spectral changes, we performed principal component analysis of the spectral dataset, focusing on the matrix content. In conclusion, Raman spectroscopy provides reliable information on chemical changes in both mineral and matrix contents, and it also identifies possible mechanisms of disuse osteoporosis.

## 1. Introduction

Osteoporosis is a disease characterized by low bone mass and the microarchitectural deterioration of bone tissue, leading to enhanced bone fragility and a consequent increase in fracture risk [1]. It is a major public health concern worldwide among elderly people and patients with primary diseases such as diabetes mellitus (DM), chronic kidney disease (CKD), and rheumatoid arthritis (RA). An increase in the number of patients with osteoporosis and associated bone fractures has resulted in significant morbidity and mortality, and hip fracture in particular not only decreases daily activity and quality of life but it also increases mortality [2,3,4].

Clinical measurements of bone strength are generally used to assess fracture risk. Bone strength is a function of bone density and bone quality [5]. Bone mineral density (BMD), which reflects bone mineral composition, accounts for 70% of bone strength [5] and it is usually determined by dual energy X-ray absorptiometry (DXA), which measures the X-ray absorption per square centimeter of bone tissue [6]. While DXA is the gold standard for the diagnosis of osteoporosis [7], it is only a moderate predictor of fracture risk [6,8,9,10]. Thus, BMD can identify people who are at increased risk of developing a fracture, but it cannot with any certainty identify individuals who will develop a future fracture [11].

Bone quality and tissue material properties are important factors in determining bone strength [12]. Bone quality refers to the bone architecture, turnover, damage accumulation, and mineralization [5]. Bone is composed of type I collagen stiffened by crystals of calcium hydroxyapatite [13]. Collagen cross-link formation affects tensile strength and the post-yield properties of bone [12]. In the last two decades, bone quality has also been used as a surrogate marker of fracture risk in osteoporosis patients. However, clinicians have no means of directly assessing bone quality. Although no single method can completely characterize bone quality, current noninvasive imaging techniques can be combined with ex vivo mechanical and compositional techniques, including peripheral quantitative CT, high-resolution MRI, micro-CT, scanning electron microscopy, and vibrational spectroscopy, to provide a comprehensive understanding of bone quality [14]. Two vibrational spectroscopic approaches, the Fourier transform infrared spectroscopy (FT-IR) and Raman spectroscopy, are promising techniques for evaluating the molecular composition of bone tissue based on interactions between photons and the molecular vibrations that occur as a result of optical absorption and scattering.

Several basic studies have evaluated bone quality using FT-IR and Raman spectroscopy [15,16,17,18,19,20,21,22,23,24,25,26]. Raman spectroscopy in particular is a minimally invasive technique and can therefore be used to assess living tissue, where it provides data on the mineral density, crystallinity, and chemical composition of intact bone tissue [16,18,19,20,21,22,23,24,25]. McCreadie et al. used Raman spectroscopy to investigate potential differences in the chemical composition of human femoral heads in 15 women with fractures and 11 women without (average age: 80.5 and 79.5, respectively), and the authors reported that the fracture cases had higher mineral/matrix ratios and higher carbonate/phosphate ratios, though the differences were not statistically significant [21]. Importantly, the prevalence of osteoporosis is known to increase significantly with age. Akkus et al. quantified age-related changes in the mineral crystallinity, degree of mineralization, and the extent of carbonate substitution in phosphate locations of rat cortical bone using Raman microspectroscopy [22]. These age-related changes are mainly influenced by hormone balance (e.g., postmenopausal osteoporosis) and physical activity (e.g., disuse osteoporosis), as well as by primary diseases such as DM, CKD, and RA [22,23]. The most popular animal model of postmenopausal osteoporosis is the ovariectomy (OVX) model. Fu et al. evaluated the mandibular cortical bone of OVX and sham-operated rats using Raman spectroscopy [24]. Changes in the mineral and organic contents were calculated as the peak parameters, and these two groups were distinguishable by principal component analysis (PCA) and linear discriminant analysis. An animal model of disuse osteoporosis, created using spinal cord injury (SCI), has also been studied using Raman spectroscopy [25].

These previous reports suggest that Raman spectroscopy can predict the risk of bone fracture. However, the mechanisms underlying decreased bone strength in osteoporosis are very complex. The pathogenesis in models of SCI-induced disuse osteoporosis is thought to be hindlimb unloading, but neural lesion effects and hormonal changes may also be involved [26]. Unilateral sciatic neurectomy (NX) is a simpler and more suitable model for understanding the bone loss induced by decreased mechanical loading [27,28,29,30]. We developed original methods to assess the molecular composition in bone tissue based on Raman spectroscopy [31,32]. In this study of NX mice, we investigated the changes in bone quality using the Raman spectroscopic technique.

## 2. Results

### 2.1. Raman Spectroscopy

Seven mice underwent unilateral sciatic neurectomy (the NX group), and three underwent sham surgery (the Sham group). Hindlimbs were harvested at 8 months after surgery, and the excised femurs and tibiae were subjected to vibrational spectroscopic and bone morphometric analyses. First, we performed Raman spectroscopic measurements of the anterior cortical surface of the proximal tibiae using an upright Raman microscope, and we compared the spectral data from the NX and Sham groups. Figure 1 presents the average spectra obtained from the NX and Sham groups. Assignments of the major peaks in Raman spectra of the bone tissue are listed in Table 1 [20,33]. The strongest band, around 960 cm^−1^, could be assigned to the ν_1_ mode of the phosphate ion (ν_1_ PO_4_^3−^) in hydroxyapatite, and ν_1_ PO_4_^3−^ corresponded to symmetric stretching of the phosphate moiety in hydroxyapatite. A single band at 960 cm^−1^ was seen in type B carbonated hydroxyapatite [34]. The carbonate ion (CO_3_^2−^) is known to occupy two different positions in the hydroxyapatite structure. It can substitute for OH^−^ groups, resulting in type A carbonated hydroxyapatite, and for PO_4_^3−^ groups, resulting in type B carbonated apatite [35]. Other phosphate bands observed at 450, 1035, and 609 cm^−1^ could be identified as the symmetric bend mode of ν_2_ PO_4_^3−^, the antisymmetric stretching mode of ν_3_ PO_4_^3−^, and the antisymmetric bend mode of ν_4_ PO_4_^3−^, respectively [36]. A prominent carbonate band (ν_1_ CO_3_^2−^) around 1070 cm^−1^, which overlapped with the ν_1_ PO_4_^3−^ band, originated from the type B carbonated hydroxyapatite. It is known that the peak intensity of the 1070 cm^−1^ band was positively correlated with the carbonate content in both the synthetic and biological samples [34]. In addition, organic matrix contents, specifically amide I (C=O stretching mode), CH_2_ deformation, amide III (C-N stretching mode), and amino acid residues (phenylalanine, proline, and hydroxyproline), were also observed at the contents respective spectra due to the abundant type I collagen in the bone matrix. The spectra of amide I at 1664 cm^−1^ and amide III around 1243–1320 cm^−1^, and the CH_2_ band at 1448 cm^−1^ and the phenylalanine band at 1003 cm^−1^, were generally observed in proteins. Proline and hydroxyproline were abundant in collagen; thus, the obtained Raman spectra accurately reflected the collagen content in the bone matrix. Nevertheless, the spectral features of the NX and Sham groups were quite similar, and it was not sufficient to compare the groups based only on the mean spectra, as shown in Figure 1.

To validate the spectral changes quantitatively, for instance, the Raman peak intensity ratios were calculated and compared between the two groups as shown in Figure 2. The ratio of the Raman peak intensity of carbonate (at 1069 cm^−1^) to that of phosphate (at 960 cm^−1^) can provide valuable insights into the chemical composition of murine or human bones, because the phosphate positions in the apatitic lattice are susceptible to ionic substitution [24]. In this study, the carbonate/phosphate ratio in NX mice was significantly higher than that in the Sham group (Figure 2a). On the other hand, to assess the relative amount of organic matrix content in bone tissue, a previous study used the matrix bands assigned mainly to the type I collagen matrix, namely the amide I band (at 1664 cm^−1^), methylene side chains (CH_2_ at 1448 cm^−1^), phenylalanine (at 1002 cm^−1^), proline (at 921 cm^−1^ and 855 cm^−1^), and hydroxyproline (at 876 cm^−1^) [19]. The mineral/matrix ratio, which indicates the degree of mineralization, corresponds to the intensity of the primary phosphate band (ν_1_ PO_4_^3−^ at 960 cm^−1^) divided by the intensity of the matrix bands (Figure 2b–d). The mineral/phenylalanine ratio and the mineral/proline + hydroxyproline ratio were significantly higher in the NX group than in the Sham group (Figure 2b,c). The mineral/CH_2_ ratio and the mineral/amide I ratio were higher in the NX group than in the Sham group, but the difference was not statistically significant (Figure 2d,e). In addition, regarding the parameters representing the matrix contents, we also investigated the amide I/CH_2_ ratio as an indicator of the amount of matrix protein versus other organic components; this ratio was lower in the NX group than in the Sham group, but the difference was not statistically significant (Figure 2f).

### 2.2. Principal Component Analysis (PCA) of the Raman Spectral Data

In addition to the quantitative analysis of the Raman spectral data, we also performed PCA to confirm the spectral differences between the NX and Sham groups. In the PCA procedure, wavenumbers (cm^−1^) of the spectral dataset were regarded as correlated variables, and the numerous variables were transformed into a few uncorrelated variables called the principal components (PCs). Figure 3 shows the score plots of the obtained PCs. First, we applied all variables to the model construction, meaning that we analyzed all the spectral regions and obtained a score plot (Figure 3a). PC1 and PC2 accounted for the most variability in the data (PC1: 68.0% and PC2: 25.5%, overall: 93.5%). Next, we selected the variables for the mineral component (Figure 3b) or the matrix component (Figure 3c). The score plots were obtained using PC2 and PC3, because PC1 reflected the variance of absolute intensity among the spectral data, which may not have contributed to validation of the spectral changes in bone quality. While the PCA result provided a good separation between the NX and Sham groups, there was still significant overlap between each other. The two groups could not be separated completely with any liner discriminant function, but the variance of each of the individual Raman spectra was visually characterized as drawn with the ovals on the score plots. In summary, multivariate PCA found that differences in both the mineral and matrix composition between the NX and Sham groups were detectable using Raman spectroscopy.

### 2.3. Bone Mineral Density

After the Raman measurements, the excised femurs and tibiae were subjected to BMD analysis using single-energy X-ray absorptiometry. Three regions of interests were defined: Proximal, midshaft, and distal. Figure 4 shows the mean BMD of the femurs and tibiae. In all regions of the femurs and tibiae, BMD was significantly decreased in the NX group compared to the Sham group.

### 2.4. Micro-CT

We also performed micro-CT analysis to observe the bone microstructure of the tibiae obtained from the NX and Sham mice. Representative three-dimensional (3D) reconstructions of the proximal tibial trabecular bone showed that NX induced bone loss and microstructural change (Figure 5). Analysis of the 3D parameters revealed that NX decreased bone volume (BV/TV), trabecular number, trabecular thickness, cancellous bone mineral density, cortical thickness, and cortical bone mineral density, and it also increased the structural model index (SMI) and trabecular separation, as shown in Figure 6. Connectivity density was lower in the NX group than in the Sham group, but the difference was not significant (Figure 6b).

## 3. Discussion

Denervation procedures, such as NX and SCI, cause osteoporotic changes in the hindlimbs in animal models. A commonly accepted pathogenic mechanism is that disuse accelerates bone resorption, especially of the cancellous bone, and consequently the bone becomes atrophic and fragile [37]. However, the pathogenesis of osteoporosis after SCI is a complex process involving not only denervation but also unloading and hormonal changes, and it is not thought to be simply considered as disuse osteoporosis [26]. In this study, we generated and analyzed a simpler model of disuse osteoporosis that induced denervation and unloading. As mentioned above, single-energy X-ray absorptiometry and micro-CT analyses confirmed that BMD was decreased in both the cancellous and cortical bone in the NX group compared to the Sham group (Figure 4, Figure 5 and Figure 6). The sciatic nerve is mainly constituted of motor and sensory nerve fibers, and mechanical unloading in the tibiae and femurs is caused by disused hindlimbs due to motor dysfunction. The sciatic nerve comprehends motor innervation lower than the knee joint, but partially in the hip joint movement, where this could be the reason why the sciatic neurectomy might affect the tibial BMD rather than the femoral BMD (Figure 4). In general, unloading inhibits bone formation and enhances bone resorption caused by osteoblastic cell suppression and osteoclastic cell activation. In terms of denervation, NX only results in the sensory denervation of bone, whereas SCI leads to sympathetic denervation, as well as sensory denervation [38]. Therefore, mechanical loading is a dominant cause of bone loss in the NX osteoporotic model but not in the SCI model.

In addition to the typical bone loss due to mechanical unloading, we also observed alterations to the bone microstructure as shown in Figure 5 and Figure 6. The bone microstructure is important for maintaining bone strength. In this study, the loss of bone strength caused numerous alterations on the micro-CT, including changes to 3D parameters; decreases in trabecular number and thickness, connectivity density, and cortical thickness; and increases in the SMI and trabecular separation. In other words, the micro-CT analysis detected reduced quality in the 3D bone microstructure in NX mice.

Vibrational spectroscopy, FT-IR, and Raman spectroscopy provide information on the molecular structure and composition of bone tissue, both of which may be related to bone quality. The results of the Raman spectroscopy in this study also suggested that reduced bone quality, characterized by changes in the bone mineral and matrix contents, might have affected bone strength in the NX mice. The intensities of Raman bands are directly proportional to the concentrations of molecules or ions that have Raman activity. According to recent articles and reviews, compositional measures of bone quality based on the Raman spectroscopy consist mainly of the mineral/matrix ratio, carbonate/phosphate ratio, collagen crosslink maturity, and crystallinity [14,19,20,39]. Appendix A presents the results of a literature survey summarizing the quantitative bone quality assessments based on Raman peak ratios. The carbonate (ν_1_ CO_3_^2−^)/phosphate (ν_1_ PO_4_^3−^) ratio was the ratio most often used ratio to assess bone mineral quality in human and animal bone tissue [21,22,23,24,25,40,41,42,43]. CO_3_^2−^ in bone mineral can modify apatite crystallinity by substitution in the apatite lattice, either to PO_4_^3−^ (major site, type B carbonate) or OH^−^ (minor site, type A carbonate) [44]. In our results, the carbonate/phosphate ratio (1069 cm^−1^/960 cm^−1^) was significantly increased in the NX mice (Figure 2a), suggesting an increased amount of the type B carbonate substituent in the cortical bone hydroxyapatite. An increased carbonate/phosphate ratio seems to correlate positively with bone fracture risk and bone aging in both humans and animals, and osteoporotic changes induced by OVX, CKD, DM, and SCI in animals, while conversely, it is negatively correlated with bone strength in exercise (Appendix A). Thus, the carbonate/phosphate ratio determined by the Raman measurement can be a reliable marker for bone mineral quality. On the other hand, the relationship between the carbonate content and mineral maturation (not aging) is still debatable. Tarnowski et al. reported that in mice, the carbonate/phosphate ratio increased with age (from postnatal to adult) [23], and it is known that in synthetic apatite, CO_3_^2−^ content increases with maturation [44,45]. Further study is needed concerning the role of carbonate content in bone mineral maturity. Like the carbonate/phosphate ratio, the mineral/matrix ratio is one of the strongest predictors of bone mechanical properties [19]. Previous studies have assessed Raman bands assigned to organic matrix content in bone, including phenylalanine (at 1002 cm^−1^), proline and hydroxyproline (922 cm^−1^ + 855 cm^−1^ + 876 cm^−1^), methylene side chains of CH_2_ (1448 cm^−1^), and amide I (1664 cm^−1^) [21,22,23,24,25,40,41,42,43,46]. The choice of matrix band is dependent on the sample preparation (e.g., fixation and embedding), instrumentation, and the band’s orientational sensitivity (or lack thereof) toward mineralized collagen fibrils [19]. In this study, we measured the intact bone tissue with ethanol fixation. It minimally affected the Raman measurement, and consequently any matrix bands that were available for the calculation of the mineral/matrix ratio. Our results showed that compared to the Sham group, the NX group demonstrated an increased degree of mineralization due to decreased bone remodeling, especially new bone formation (Figure 2). Bone remodeling involves the resorption of old bone by osteoclasts and its subsequent replacement with new bone formed by osteoblasts. With increased bone remodeling, the amount of young bone, which contains relatively less mineral content than older bone, generally increases under healthy conditions. In the process of bone development and growth in mice, the mineral/matrix ratio drastically increases [23]. In osteoporotic bone, abnormal bone remodeling results in controversial behaviors of the mineral/matrix ratios on Raman spectroscopy, as shown in Appendix A. The mineral/matrix ratio was higher in human osteoporosis, aged bone in humans and mice, and systemic disease (DM and CKD) model rats, and it was lower in SCI and OVX rats. This suggested that imbalances in the mineral and matrix content may be associated with bone fragility. In the case of rat OVX, as reported by Fu et al. [24], the lower mineral/matrix ratio is likely to be consistent with the increased remodeling, especially enhanced bone resorption, that is induced by estrogen deficiency. In this study, the amide I/methylene side chain ratio (1664 cm^−1^/1448 cm^−1^) was decreased in the NX, and whilst this change was not statistically significant, this parameter may reflect compositional changes, excluding those involving mineral content, in the bone matrix. The amide I band is specific to proteins, and the CH_2_ band arises from organic components, such as proteins, lipids, and sugars. Therefore, a decreased amide I/CH_2_ ratio could result from degradation of the collagen matrix by the matrix metalloproteinase activity of osteoclasts.

Reduced bone quality induced by NX was quantitatively characterized by the Raman band ratios of carbonate/phosphate, mineral/matrix, and amide I/CH_2_, as discussed above. Other combinations of the Raman/FT-IR bands originating from the matrix components (e.g., 1660 cm^−1^/1690 cm^−1^ on the amide I band) have been also employed for estimating collagen maturity in previous reports [15,41,43,47,48]. Under physiological conditions, collagen fibrils are crosslinked with pyridinoline (mature crosslinks) and dihydroxy lysine-norleucine (immature crosslinks) during matrix maturation. However, regarding the Raman peak assignment for collagen crosslinking, we considered the ratio of 1660 cm^−1^ to 1690 cm^−1^ to be a potential metric for the amount of nonreducible/reducible crosslinking of collagen [49]. Nevertheless, the intermolecular crosslinks in collagen maintain the mechanical properties of bone. To elucidate whether the amide I band is sensitive to the maturation of collagen crosslinking in the bone matrix, further chemical studies should be performed. Instead, in this study, we performed further analysis based on the Raman spectroscopic data to find possible markers that indicated the state of collagen crosslinking and any other changes in the bone matrix of the NX mice. The results of the PCA showed that the NX and Sham groups were distinguishable based on the overall spectral data, as well as the data for the mineral and matrix bands (Figure 3). The mineral bands used for the PCA, which were mainly assigned to carbonate and phosphate content, may have contributed to the dispersion of the scatter plots based on the PCs (Figure 3b). On the other hand, the matrix bands in the PCA analysis, including for amide I (1630–1700 cm^−1^), methylene side chains (1420–1480 cm^−1^), amide III (1200–1300 cm^−1^), and proline and hydroxyproline (800–900 cm^−1^), were also changed in the NX group (Figure 3c). The dispersion patterns in the scatter plots for mineral content (Figure 3b) and matrix content (Figure 3c) were different from each other, but they suggested that the Raman spectral changes in the matrix content also characterized the reduced bone quality in NX mice. On the other hand, the significant overlap between the sham and NX group still remains. The overlap may possibly derive from the compositional differences at different points rather than the individual differences. The Raman spectroscopic approach with PCA has the potential to comprehensively and nondestructively elucidate the chemical alterations in bone tissue, because the acquired Raman spectra are multivariate data that include information on the molecular composition of the mineral and matrix content. Thus, Raman spectroscopy is a promising validation and diagnostic tool in osteoporosis.

## 4. Materials and Methods

### 4.1. Animals and Surgical Procedures

In this study, we generated unilaterally sciatic neurectomized mice. Ten male C57/BL6J mice (aged 5 weeks) were purchased from CLEA Japan (Tokyo, Japan). All the mice were fed commercial food and water ad libitum and were maintained under controlled lighting conditions (12-h/12-h light/dark cycle) throughout the study. The mice were allowed to acclimatize for 6 weeks before the experiment began. Sciatic neurectomy was performed to induce bone loss with unloading. Seven mice were randomly assigned to the NX group. The mice were anesthetized by isoflurane. The hindlimb was shaved and sterilized. The skin incision was made on the posterior to the femoral trochanteric region. The sciatic nerve was exposed and a 1-cm-long segment was resected, as described in Reference [30]. The muscular branch was also resected. Sham operations were performed in three mice (Sham group). The sham group mice were anesthetized and incised in the same manner as the NX group, but no sciatic neurectomy was done. Eight months after surgery, all the mice were sacrificed and the hindlimbs were collected and fixed with 70% (*v*/*v*) ethanol. All the experimental animal protocols in this study were approved by the Ethical Committee for Animal Experiments of Ehime University (#05NU74-1).

### 4.2. Raman Spectroscopy

Raman spectra were obtained with an inVia Raman Microscope (Renishaw, UK). Raman spectra were randomly collected on the anterior cortical surface of the proximal tibia at 5 points in each bone. The excitation laser (785 nm, 10 mW) was focused on the specimen via a 60× water-immersion objective lens (Olympus LUMPLFLN, 60 × W, NA = 1.0). The exposure time was 30 s, the acquisition number was two for each measurement, and the grating had 1200 lines/mm. The spectral range was 300–1800 cm^−1^, and the spectral resolution was approximately 1 cm^−1^. Baseline correction of the obtained Raman spectral data was performed by quartic polynomial curve fitting with MATLAB (MathWorks, Natick, MA, USA) prior to quantitative and statistical analyses. We identified the specific Raman peaks that could be assigned to the vibrational modes of molecular composition in the bone tissue. The Raman peak assignments for hydroxyproline (876 cm^−1^), proline (922 cm^−1^ and 855 cm^−1^), phosphate ν_1_ (960 cm^−1^), phenylalanine (1002 cm^−1^), carbonate (1069cm^−1^), CH_2_ (1448 cm^−1^), and amide I (1664 cm^−1^) have been reported in past studies, as in References [20,33] (Table 1). Ratios of carbonate/phosphate, phosphate ν_1_/phenylalanine, phosphate ν_1_/proline+hydroxyproline, phosphate ν_1_/CH_2_, phosphate ν_1_/amide I, and amide I/CH_2_ were calculated with each of the intensities at the peak top for quantitative analysis.

### 4.3. Multivariate Analysis of Raman Spectral Data

For multivariate analysis, a Raman spectral dataset was prepared for PCA. Raman spectra, which were obtained from multiple acquisition at different spots in the same specimen, were averaged as an individual to compensate for the site dependency. Sirius 7.1 software (Pattern Recognition Systems, Bergen, Norway) was employed for the PCA. In the PCA procedure, wavenumbers (cm^−1^) of the spectral dataset were regarded as variables that were correlated with each other, and the data were mean-centered. The variables were transformed into a few uncorrelated variables or PCs. Discrimination models were built based on all the spectral regions, the partially extracted spectral regions for mineral components (400–500, 550–630, 900–980, and 1030–1100 cm^−1^), and the partially extracted spectral regions for matrix components (800–900, 1200–1300, 1420–1480, and 1630–1700 cm^−1^).

### 4.4. BMD Measurement

BMD was measured by single-energy X-ray absorptiometry with a bone mineral analyzer (DCS-600EX-III; ALOKA, Tokyo, Japan). The excised femurs and tibiae were subjected to BMD measurements. Three regions of interest were defined: Proximal, midshaft, and distal.

### 4.5. Micro-CT Scanning

Micro-CT analysis was performed as previously reported in Reference [50]. Tibiae were fixed with 70% (*v*/*v*) ethanol and subjected to micro-CT analysis using a Scanco Medical μCT35 System (Scanco Medical, Brüttisellen, Switzerland), with an isotropic voxel size of 6 μm for trabecular analyses according to the guidelines of the American Society for Bone and Mineral Research (ASBMR) [51]. A total of 350 slices of proximal tibial metaphysis starting at 0.6 mm from the end of the growth plate were scanned and analyzed. Three-dimensional reconstructions were generated and analyzed according to the ASBMR guidelines [51].

### 4.6. Statistics

Data were presented as the average ± the standard error of the mean. Significance was determined using the Student’s *t* test. Values of *p* < 0.05 were considered statistically significant.

## Figures and Tables

**Figure 1 molecules-23-03081-f001:**
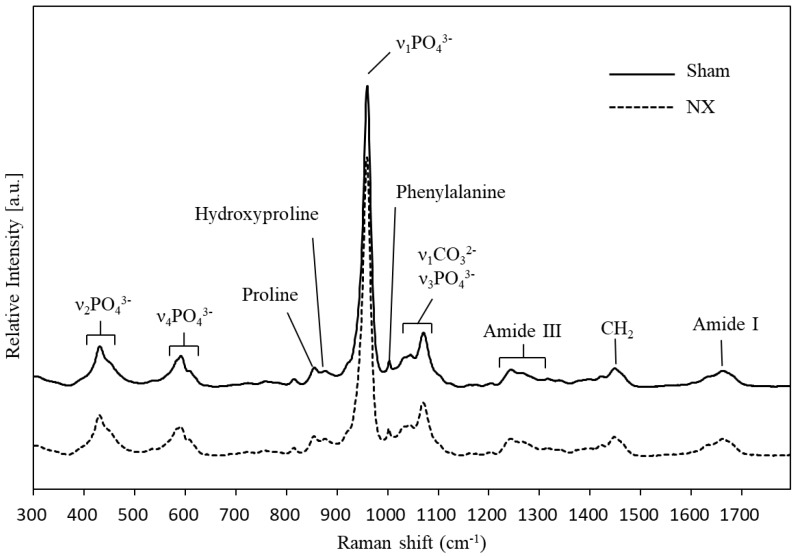
The average Raman spectra obtained from neurectomized (NX group) mice and Sham operation mice (Sham group). Previous studies can be used to assign apparent peaks in Raman spectra originating from the bone mineral and matrix components (Table 1). Raman peaks originating from the mineral components are as follows: ν_2_ PO_4_^3−^, 430–450 cm^−1^; ν_4_ PO_4_^3−^, 584–610 cm^−1^; ν_1_ PO_4_^3−^, 960 cm^−1^; and ν_1_ CO_3_^2−^ ν_3_ PO_4_^3−^, 1035–1076 cm^−1^. Those originating from the matrix components are as follows: Proline, 855 cm^−1^ and 922 cm^−1^; hydroxyproline, 876 cm^−1^; phenylalanine, 1002 cm^−1^; amide III, 1243–1320 cm^−1^; CH_2_, 1448 cm^−1^; and amide I, 1664 cm^−1^.

**Figure 2 molecules-23-03081-f002:**
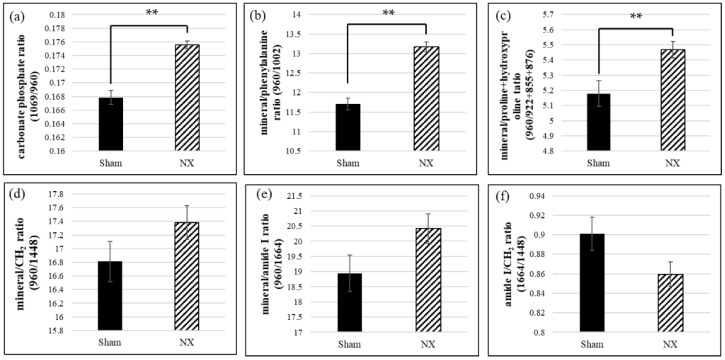
Raman peak intensity ratios in the NX and Sham groups. (**a**) Carbonate/phosphate ratio (1069 cm^-1^/960 cm^−1^), (**b**) mineral/phenylalanine ratio (960 cm^−1^/1002 cm^−1^), and (**c**) mineral/proline+hydroxyproline ratio (960 cm^−1^/922 cm^−1^ + 855 cm^−1^ + 876 cm^-1^) were significantly higher in the NX group than in the Sham group. (**d**) Mineral/CH_2_ ratio (960 cm^−1^/1448 cm^−1^) and (**e**) mineral/amide I ratio (960 cm^−1^/1664 cm^−1^) were also higher in the NX group, and (**f**) amide I/CH_2_ ratio (1664 cm^−1^/1448 cm^−1^) was lower in the NX group, although there were no significant differences. ** *p* < 0.01.

**Figure 3 molecules-23-03081-f003:**
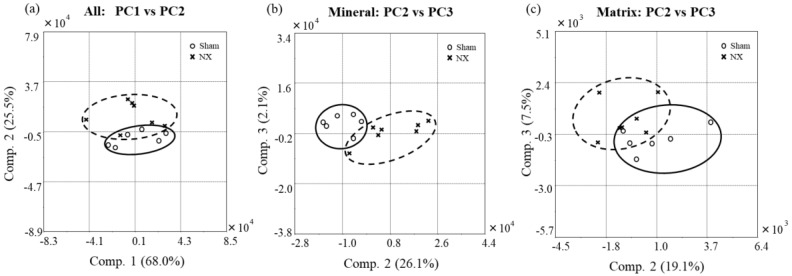
Score plots of the Principal Component Analysis (PCA) analysis of Raman spectra. The X and Y axes include any principal components (PCs) that differed between the NX and Sham groups, with the distance on the axis indicating the degree of difference. PCA plots were built with (**a**) all spectral regions, (**b**) a partially extracted spectral region for the mineral component, and (**c**) a partially extracted spectral region for the matrix component.

**Figure 4 molecules-23-03081-f004:**
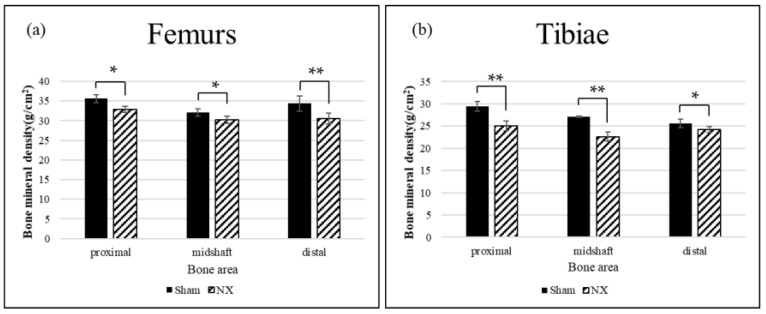
Mean bone mineral density (BMD) of the (**a**) femurs and (**b**) tibiae. The BMD of both the femurs and tibiae was significantly decreased in the NX group compared to the Sham group. * *p* < 0.05, ** *p* < 0.01.

**Figure 5 molecules-23-03081-f005:**
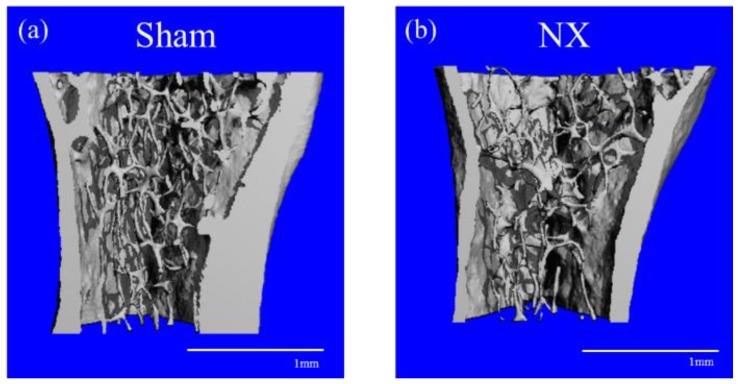
Volume rendering images constructed from X-ray micro-CT data of the proximal tibiae of (**a**) Sham and (**b**) NX mice. In the NX mouse, the thickness of the cortical bone is obviously decreased, and the microstructure of the cancellous bone is altered.

**Figure 6 molecules-23-03081-f006:**
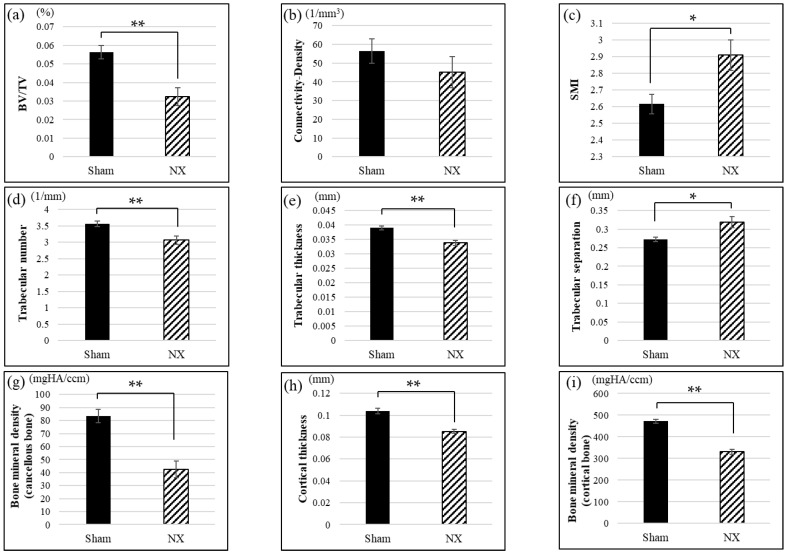
Micro-CT parameters. Relative to the Sham group, the NX group demonstrated an altered (**a**) bone volume (BV/TV), (**b**) connectivity density, (**c**) structural model index (SMI), (**d**) trabecular number, (**e**) trabecular thickness, (**f**) trabecular separation, (**g**) cancellous BMD, (**h**) cortical thickness, and (**i**) cortical BMD. * *p* < 0.05, ** *p* < 0.01.

**Table 1 molecules-23-03081-t001:** Assignment of Raman peaks in bone tissue, as in References [20,33].

Raman Shift, cm^−1^	Assignment	Comments
430	ν_2_ PO_4_^3−^	Strong band
450	ν_2_ PO_4_^3−^	Shoulder on 430 cm^−1^ band
584–590	ν_4_ PO_4_^3−^	Multiple partially resolved components
609	ν_4_ PO_4_^3−^	Shoulder on 590 cm^−1^ band
853	ν(C-C)	Collagen proline, may include δ(C-C-H) contribution from tyrosine
872	ν(C-C)	Mostly collagen hydroxyproline
955	ν_1_ PO_4_^3−^	Transient bone mineral (P-O) phase, usually seen in immature bone
957	ν_1_ PO_4_^3−^	Bone mineral containing extensive HPO_4_^2−^, usually immature
959–962	ν_1_ PO_4_^3−^	Bone mineral, mature
1003	ν(C-C)	Phenylalanine
1035	ν_3_ PO_4_^3−^	Overlaps with proline ν(C-C) component
1048	ν_3_ PO_4_^3−^	
1070	ν_1_ CO_3_^2−^	Overlaps with component of ν_3_ PO_4_^3−^
1076	ν_3_ PO_4_^3−^	Overlaps with component of ν_1_ CO_3_^2−^
1242	Amide III	Protein β-sheet and random coils
1272	Amide III	Protein α-helix
1340	Amide III	Protein α-helix, sometimes called CH_2_CH_2_ wag
1446	δ(CH_2_)	Protein CH_2_ deformation
1660	Amide I	Strongest amide I ν(C=O) component, polarization sensitive
1690	Amide I	Shoulder, prominent with immature cross-links
		The band also relates to β-sheet or disordered secondary structure

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
