# Peer review of "Raman Spectroscopic Analysis to Detect Reduced Bone Quality after Sciatic Neurectomy in Mice"

_molecules, 2018, doi:10.3390/molecules23123081_

Round 1
Reviewer 1 Report
The present study talk about tracking changes in the bone composition by Raman after sciatic nerve destruction. The study could be of interest to the journal. Experiments are sound and results are supporting the hypothesis. Paper can be accepted with minor revisions. 1. My major concern is with the way authors have recorded Raman spectrum. With the microscope with a 60X objective spot size will be in microns. Authors recorded spectra at random points, which is not justified. With such a small spot random spots won't reflect the real changes in the bone. That's the reason why Raman difference between Sham and NX mice are not big as it is reflected by other techniques. Ideal way of recording spectra will be to acquire spectra at same spots in all specimens and compare them. 2. Authors should discuss clearly why differences after BMD experiments are more in tibia as compare to femur. 3. PCA results also suggest same, there is significant overlap between sham and NX group, mainly because of differences in mineral composition at different points. 4. How did authors calculate the area/intensity of Raman bands. Just the integration of the intensity or authors did some curve fitting. 5. Ideally authors should have include one more group of control (normal) mice, that would provide information about differences in Sham control.Author Response
We appreciate the time and effort of the referees and the editor to improve our manuscript. First of all, we have revised our manuscript according to the referee’s suggestion. Our point-by-point responses to the comments are presented below.
1. My major concern is with the way authors have recorded Raman spectrum. With the microscope with a 60X objective spot size will be in microns. Authors recorded spectra at random points, which is not justified. With such a small spot random spots won't reflect the real changes in the bone. That's the reason why Raman difference between Sham and NX mice are not big as it is reflected by other techniques. Ideal way of recording spectra will be to acquire spectra at same spots in all specimens and compare them.
As the reviewer pointed out, the excitation laser spot is too small to evaluate the compositional changes in the tibial cortical bone. In this study, the small spot acquisitions were randomly performed at five measurements per bone to compensate the sitedependency. In addition, we chose the anterior part of proximal tibia as an optimal measurement target which is easily recognizable under microscopy for each bone. Regarding the quantitative analysis based on Raman peak intensity rations, our strategy allows us to capture the real changes in the bone. Taking the reviewers opinion into account, we added the following sentences into the materials and methods section and the discussion section in the revised manuscript.
Page 10, line 357 ... was prepared for PCA. Raman spectra, which were obtained from multiple acquisition at different spots in the same specimen, were averaged as an individual to compensate the sitedependency. Sirius 7.1 software …
Page 9, line 315 …bone quality in NX mice. On the other hand, the significant overlap between sham and NX group still remains. The overlap may possibly derive from the compositional differences at different points rather than the individual differences. The Raman spectroscopic approach…
2. Authors should discuss clearly why differences after BMD experiments are more in tibia as compare to femur.
Thank you very much for your suggestion. We added the interpretation for that into the discussion part as follows.
Page 8, line 229 … motor dysfunction. Sciatic nerve comprehends motor innervation lower than the knee joint, but partially in the hip joint movement, this could be the reason why the sciatic neurectomy might affect the tibial BMD rather than femoral BMD (Fig. 4). In general, …
3. PCA results also suggest same, there is significant overlap between sham and NX group, mainly because of differences in mineral composition at different points.
As mentioned in the first comment, the measurement-site-dependent bias could be avoided by multiple acquisition and averaging process. However, as the reviewer concerned, the overlap may possibly derive from the compositional differences at different points rather than the individual differences. In this point, we have addressed in Question 1.
4. How did authors calculate the area/intensity of Raman bands. Just the integration of the intensity or authors did some curve fitting.
We calculated the Raman ratio using just the peak intensity at the peak top without any curve fitting. According to the comment, we described it in the revised manuscript as follows.
Page 10, line 353 …phosphate ν1/CH2, phosphate ν1/amide I, and amide I/CH2 were calculated with each of the intensities at the peak top for quantitative analysis.
5. Ideally authors should have include one more group of control (normal) mice, that would provide information about differences in Sham control.
As the reviewer pointed out, sham surgery may possibly affect the results due to the tissue damage. We are planning to design of future studies including normal group in addition to sham group.
Reviewer 2 Report
In this manuscript, the authors use Raman spectroscopy to examine bone quality after sciatic neurectomy. They present interesting results on the quality of bone with previously established Raman markers for bone quality, crystallinity, etc. While the results are interesting, there are some issues with the manuscript, outlined below, which would have to be addressed before it can be considered for publication:
The main issue with the manuscript is the interpretation of the PCA results. While figure 2 shows clear differences between the sham and NX groups, the same cannot be claimed for the PCA results in Figure 3. For all 3 PC plots, there is not a significant separation between the two groups and there is a good amount of overlap between them. While the ovals drawn around the PC scores draw the eye to the two groups, this does not automatically make it possible to claim separation. The authors need to re-address their treatment of these results, especially on page 5, lines 178-179, where there is a definitive statement what the PCA results show that is not as straightforward as the authors state.
Additionally, the quality of Figure 3 should be improved. The figure can be made bigger, and if not, the fonts should be increased so that the reader can actually read the labels and differentiate between the markers. It might be easier to not label each marker but instead have distinctive shapes to differentiate between them (or use different colors)
While the authors describe the advantages of using neurectomy as a method for inducing bone loss in the methods section, it would be helpful to the reader to have this discussed earlier in the manuscript, specifically at page 2, lines 95-96.
Table 2 has a lot of data in and and is not very easy to understand. The authors should look at improving how this information is presented. Alternatively, this could go into a supplemental information section.
Author Response
We appreciate the time and effort of the referees and the editor to improve our manuscript. First of all, we have revised our manuscript according to the referee’s suggestion. Our point-by-point responses to the comments are presented below.
The main issue with the manuscript is the interpretation of the PCA results. While figure 2 shows clear differences between the sham and NX groups, the same cannot be claimed for the PCA results in Figure 3. For all 3 PC plots, there is not a significant separation between the two groups and there is a good amount of overlap between them. While the ovals drawn around the PC scores draw the eye to the two groups, this does not automatically make it possible to claim separation. The authors need to re-address their treatment of these results, especially on page 5, lines 178-179, where there is a definitive statement what the PCA results show that is not as straightforward as the authors state.
Thank you very much for your fruitful suggestion. As the reviewer pointed out, there is significant overlap between sham and NX group. The overlap may possibly derive from the compositional differences at different points rather than the individual differences. According to the reviewer’s comment, we added the following statements in the results part and the discussion part.
Page 5, line 179 …spectral changes in bone quality. While the PCA result provided good separation between NX and Sham groups, there is still significant overlap between each other. Those two groups could not be separated completely with any liner discriminant function, but the variance each of the individual Raman spectra was visually characterized as drawn with the ovals on the score plots. In summary, multivariate PCA found that…
Page 9, line 315 …bone quality in NX mice. On the other hand, the significant overlap between sham and NX group still remains. The overlap may possibly derive from the compositional differences at different points rather than the individual differences. The Raman spectroscopic approach…
Additionally, the quality of Figure 3 should be improved. The figure can be made bigger, and if not, the fonts should be increased so that the reader can actually read the labels and differentiate between the markers. It might be easier to not label each marker but instead have distinctive shapes to differentiate between them (or use different colors)
Thank you very much for your comment. We modified Figure 3.
While the authors describe the advantages of using neurectomy as a method for inducing bone loss in the methods section, it would be helpful to the reader to have this discussed earlier in the manuscript, specifically at page 2, lines 95-96.
In the manuscript, we have mentioned earlier about the reason why we chose neurectomy as an osteoporotic model in the introduction section in brief. After that, in the discussion part, more detail focusing of the mechanism of bone loss induced by the mechanical unloading in comparison to another method, spinal cord injury prior to description in the methods section.
Table 2 has a lot of data in and is not very easy to understand. The authors should look at improving how this information is presented. Alternatively, this could go into a supplemental information section.
Thank you very much for your suggestion. Table 2 moved to a supplemental material.
Reviewer 3 Report
The manuscript by Ishimaru et al shows the capabilities of Raman spectroscopy of bone tissue as tool to predict bone fracture potential, as compared to golden standard techniques currently in use. The study is extremely well carried out and presented, and it represents a good step forward in understanding the molecular insights into bone fragility. The conclusions are supported by the data, and all experimental procedures are well described.
This reviewer noticed that the experiments described herein have been previously published in two conference proceedings (albeit in a much more concise way, excluding several details and most of the discussion included in the present manuscript). As per the “Instructions for Authors” (https://www.mdpi.com/journal/molecules/instructions#preprints), Molecules allows publication of conference proceedings papers under a set of conditions: “Expanded and high quality conference papers can be considered as articles if they fulfil the following requirements: (1) the paper should be expanded to the size of a research article; (2) the conference paper should be cited and noted on the first page of the paper; (3) if the authors do not hold the copyright of the published conference paper, authors should seek the appropriate permission from the copyright holder; (4) authors are asked to disclose that it is conference paper in their cover letter and include a statement on what has been changed compared to the original conference paper. Molecules does not publish pilot studies or studies with inadequate statistical power.” It is this reviewer’s opinion that condition number 1 has been satisfied, however condition number 2 has not (conditions 3 and 4 are not easily verifiable by the reviewer), although if the editor considers the manuscript to be acceptable it would be a minor revision to add said disclosure and citation. The two conference proceedings are: Ishimaru, Y. Et al, Eds.; SPIE, 2017; Vol. 10251, p 102511D and Ishimaru, Y. et al., Proc SPIE 10497, 1049722 (2018).
A few details regarding the procedures related to the “Sham” samples should be added: were the mice anesthetized, incised, but no sciatic neurectomy was performed?
Following the above consideration, I would recommend acceptance after minor revision.
Author Response
We appreciate the time and effort of the referees and the editor to improve our manuscript. First of all, we have revised our manuscript according to the referee’s suggestion. Our point-by-point responses to the comments are presented below.
This reviewer noticed that the experiments described herein have been previously published in two conference proceedings (albeit in a much more concise way, excluding several details and most of the discussion included in the present manuscript). As per the “Instructions for Authors” (https://www.mdpi.com/journal/molecules/instructions#preprints), Molecules allows publication of conference proceedings papers under a set of conditions: “Expanded and high quality conference papers can be considered as articles if they fulfil the following requirements: (1) the paper should be expanded to the size of a research article; (2) the conference paper should be cited and noted on the first page of the paper; (3) if the authors do not hold the copyright of the published conference paper, authors should seek the appropriate permission from the copyright holder; (4) authors are asked to disclose that it is conference paper in their cover letter and include a statement on what has been changed compared to the original conference paper. Molecules does not publish pilot studies or studies with inadequate statistical power.” It is this reviewer’s opinion that condition number 1 has been satisfied, however condition number 2 has not (conditions 3 and 4 are not easily verifiable by the reviewer), although if the editor considers the manuscript to be acceptable it would be a minor revision to add said disclosure and citation.
Thank you very much for your comment. We mentioned in the manuscript as follows;
Page 2, Line 96 Unilateral sciatic neurectomy (NX) is a simpler and more suitable model for understanding bone loss induced by decreased mechanical loading [27-30]. We have been developed our original methods to assess the molecular composition in bone tissue based on Raman spectroscopy ever before [31, 32]. In this study of NX mice, we investigated changes in bone quality using the Raman spectroscopic technique.
We added the two conference proceedings into the references.
[31] Ishimaru, Y.; Oshima, Y.; Imai, Y.; Iimura, T.; Takanezawa, S.; Hino, K.; Miura, H., Detection of changes in bone quality of osteoporotic model induced by sciatic nerve resection by using Raman spectroscopy. Proc. SPIE 10497, San Francisco, USA, 30 January 2018, Daniel L. Farkas; Dan V. Nicolau; Robert C. Leif; SPIE: Bellingham WA USA
[32] Ishimaru, Y.; Oshima, Y.; Imai, Y.; Iimura, T.; Takanezawa, S.; Hino, K.; Miura, H., Evaluation of bone quality in osteoporosis model mice by Raman spectroscopy. Proc. SPIE 10251, Yokohama, Japan, 21 April 2017; Toyohiko Yatagai; Yoshihisa Aizu; Osamu Matoba; Yasuhiro Awatsuji; SPIE: Bellingham WA USA
A few details regarding the procedures related to the “Sham” samples should be added: were the mice anesthetized, incised, but no sciatic neurectomy was performed?
Thank you very much for your suggestion. We added the interpretation in the revised manuscript as follows;
Page 10, Line 333 …Sham operations were performed in three mice (Sham group). Sham group were anesthetized and incised in the same manner as NX group, but no sciatic neurectomy was done. Eight months after surgery, …
Reviewer 4 Report
In this manuscript, Raman spectroscopy is used to study the changes in bone quality of mice..
Sciatic neurectomy (NX) was performed in male C57/BL6J mice (NX group) as a model of disuse 29 osteoporosis, and sham surgery was used as an experimental control (Sham group). Quantitative analysis based on Raman peak intensities showed that the carbonate/phosphate ratio and mineral/matrix ratio were significantly higher in the NX group than that one in the Sham group. Bone mineral density (BMD) measurement and micro-CT measurement are performed to investigate the pathogenesis of osteoporosis. Direct evidence of alterations in mineral content associated with mechanical properties of bone is observed. Reduced bone quality induced by NX was quantitatively characterized by Raman band ratios of carbonate/phosphate, mineral/matrix, and amide I/CH2.Principal component analysis of the spectra supports this conclusion. Raman spectroscopy was shown to provide reliable information on chemical changes in both mineral and matrix contents, and also identifies possible mechanisms of disuse osteoporosis.
The experiment is performed at the high technical level. The manuscript is well written, logically organized and properly divided to several sections.
Author Response
We appreciate the time and effort of the referees and the editor to improve our manuscript. We have revised our manuscript according to the other referee’s suggestion.
Round 2
Reviewer 2 Report
The authors have addressed comments from the previous review. The English for the added sentences need to be edit.